# Interventional Radiology in the Treatment of Pancreatic Adenocarcinoma: Present and Future Perspectives

**DOI:** 10.3390/life13030835

**Published:** 2023-03-20

**Authors:** Ernesto Punzi, Claudio Carrubba, Andrea Contegiacomo, Alessandro Posa, Pierluigi Barbieri, Davide De Leoni, Giulia Mazza, Alessandro Tanzilli, Alessandro Cina, Luigi Natale, Evis Sala, Roberto Iezzi

**Affiliations:** 1Dipartimento di Diagnostica per Immagini, Radioterapia Oncologica ed Ematologia-Istituto di Radiologia, Fondazione Policlinico Universitario A. Gemelli IRCCS, l.go A gemelli 8, 00168 Rome, Italy; 2Istituto di Radiodiagnostica, Università Cattolica del Sacro Cuore, 00168 Rome, Italy

**Keywords:** pancreatic adenocarcinoma, thermal ablation, irreversible electroporation, liver metastases chemoembolization, immunotherapy

## Abstract

Pancreatic ductal adenocarcinoma (PDAC) is a lethal disease; patients’ long-term survival is strictly linked to the surgical resection of the tumor but only a minority of patients (2–3%) have a resectable disease at diagnosis. In patients with surgically unresectable disease, interventional radiology is taking on an increasing role in treatment with the application of loco-regional percutaneous therapies. The primary purposes of this narrative review are to analyze the safety and efficacy of ablative techniques in the management of borderline resectable and locally advanced diseases and to underline the role of the interventional radiologist in the management of patients with distant metastases. The secondary purpose is to focus on the synergy between immunotherapy and ablative therapies.

## 1. Background

Pancreatic ductal adenocarcinoma (PDAC) remains a lethal disease, with a 5-year relative survival of 10.8%, and a complex treatment approach [1]. The vast majority of PDACs involve the head of the organ, while 20–25% are in the body/tail [1]; patients with tumors of the head present with jaundice, generally leading to an earlier diagnosis than the latter group. The location of the tumor also influences the type of surgery the patient undergoes (duodenocephalopancreasectomy in case of head cancer vs. splenopancreasectomy in case of tail and body cancers).

Interventional radiology (IR) has a fundamental role in diagnostic and preoperative phases; in patients who are not eligible for ecoendoscopic biopsy, a combination of computed tomography (CT) and/or ultrasound (US) guidance may be used as guide for percutaneous biopsy. Moreover, when there is not a safe route for a direct percutaneous biopsy, a cytologic brushing of pancreatic stenosis through a percutaneous access to the biliary tract can be used. In patients with jaundice, surgery increases the risk of postoperative complications; as such, preoperative biliary drainage improves the postoperative outcome [2].

Patients with PDAC are usually divided into four categories, according to the extent of the disease: resectable, borderline resectable, locally advanced, and metastatic (Table 1) [3]. Even though systemic therapies have improved in their effectiveness over the years, long-term survival is strictly linked to the surgical resection of the tumor [3,4,5]. Unfortunately, only a minority of patients (2–3%) [6] have a surgically resectable disease at diagnosis (pancreaticoduodenectomy for head tumors and distal pancreatectomy for tumors of the body/tail); this is in accordance with the fact that PDAC often causes few or no symptoms before it develops to an advanced stage [7]. In the remaining percentage of patients, interventional radiology is taking on an increasing role with percutaneous techniques in loco-regional disease and with both percutaneous and endovascular procedures in patients with metastatic disease.

Image-guided ablation therapies are indicated in patients with locally advanced pancreatic cancer (LAPC) who do not respond to neoadjuvant chemoradiotherapy and in patients unfit for surgery [8]. They include thermal (radiofrequency ablation, RFA; microwave ablation, MWA; and cryoablation, CA) and non-thermal ablation techniques (irreversible electroporation; IRE) [9,10]. On the other hand, both percutaneous and endovascular approaches can be used in the treatment of liver metastases.

All these procedures induce antigens release due to tumor necrosis, making them particularly useful as an adjunct to immunotherapy.

On this basis, the purpose of this narrative review is to analyze the role of interventional radiology in the management of loco-regional and metastatic diseases and to underline the synergic effect of ablative therapies and immunotherapy.

## 2. Literature Search Strategy and Information Sources

Once the review query was defined, a literature search was performed using three electronic bibliographic databases (PubMed, BioMed Central, and Scopus) seeking original studies assessing the application of interventional radiology procedures in the treatment of locally advanced pancreatic cancer. The studies were identified using the following medical subject headings (MeSH) and keywords including: “cryoablation”, “radiofrequency”, “microwave ablation”, “thermal ablation”, “irreversible electroporation“, “chemoembolization”, “immunotherapy”, “pancreatic cancer”, and “pancreatic liver metastases”. The search was restricted to the English language. We analyzed clinical studies only as full texts. Conference papers, surveys, letters, editorials, book chapters, and reviews were excluded. Two independent authors (EP, CC) screened citations in titles and abstracts to identify appropriate papers.

## 3. Loco-Regional Therapies

### 3.1. Thermal Ablation

Thermal ablation technologies exploit high (RFA, MWA) or low temperatures (cryoablation) to achieve tumor ablation.

Technically, the ablation procedure consists of the placement of one or more needles into the tumor; it may be performed during surgery or percutaneously, under CT or US guidance. US-guidance allows real-time observation of the thermal damage [11].

The primary limitation of these techniques is the proximity of the pancreas to several vital structures, which puts them at risk of thermal injury and makes complete ablation difficult to achieve. Pancreatitis, pancreatic fistula, perforation of hollow viscera, and pseudoaneurysms of the near vessels are the most common complications, but the types of complications are widely varied and included portal vein thrombosis, peritoneal cavity abscess and abdominal fluid collection, transient ascites, hemoperitoneum, acute renal failure, hepatic insufficiency, pseudomembranous colitis, gastric bypass fistula, gastric ulcer, choledocholithiasis, and pneumonia [12,13].

#### 3.1.1. Radiofrequency Ablation

RFA produces tissue coagulative necrosis through high local temperatures generated by a high frequency alternating current [14].

The ablation parameters, such as current (Amperes) and the time of ablation (s), are adjusted according to the size and morphology of the lesion and to the tissue impedance, which is recorded by the needle tip [15]. The ideal target temperature is between 90 °C and 100 °C.

This technique has been widely used in many solid organ malignancies, including hepatocellular carcinoma and lung tumors [16,17]. In pancreatic adenocarcinoma (PA) patients, the ideal target lesion is less than 3.5 cm in size and located far from large vessels to avoid excessive heat dissipation (heat-sink effect) [11,14,15]. On the other hand, a safety margin of 5 mm from adjacent vital anatomic structures is ideal in order to prevent thermal injuries [15]. However, a reduction of the heat sink effect may be obtained by a bipolar system of RFA, lowering the risk of pancreatic and peripancreatic thermal injury [18].

No randomized controlled trials regarding RFA effectiveness have been performed [11]. However, a nonrandomized study by Giardino et al. showed an overall survival (OS) up to 25.6 mo after RFA for LAPC [19]. Paiella et al. [20] reported a median OS of 30 mo for patients subjected to RFA and a median OS of 25.6 mo in the patients subjected to primary treatments plus RFA plus further systemic treatments. Furthermore, Girelli et al. reported a 1-year OS of 41% in a prospective study of 100 cases treated with RFA [21].

The rates of RFA-related overall complications ranged from 10% to 43%, while the rates of mortality ranged from 0% to 19% [22]; these were generally due to gastrointestinal bleeding and sepsis. In accordance with these data, Girelli et al. reported that the overall morbidity was approximately 26%, while the mortality rate was 1.8% [21].

#### 3.1.2. Microwave Ablation

Electromagnetic microwaves heat biological tissues through the dielectric effect; microwaves agitate water molecules, producing friction and heat, and thus inducing cellular death via coagulation necrosis.

MWA has several supposed improvements over RFA, including consistently higher intratumoral temperatures, larger ablation volumes, faster ablation times, and less sensitivity to the heat-sink effect [23].

Regarding the safety and efficacy of MWA in LAPC, only four studies have been published [24,25,26,27] and all of them suffer from limited follow-up and lack of survival data.

Vogl et al. presented the most numerous population (20 patients); they reported 100% technical success, absence of major complications, and 9.8% minor complications. The primary limitations of their study are that 3-mo follow-up imaging was available in only half of the patients (10 patients) and no OS was reported [24].

Lygidakis et al. [25] enrolled 15 patients with large pancreatic tumors (average tumor size of 6 cm). Due to the tumor size, only partial ablation was achieved in all patients; however, no major complications were reported. The longest follow-up was 22 months, but survival data of the study’s population was not published.

In the experience of Carrafiello et al. [26], five patients with LAPC underwent percutaneous MWA and five patients during open surgery. As major complications, one pancreatic pseudocyst requiring drainage and one arterial pseudoaneurysm were reported. The 1-year survival rate was 80%.

Ierardi reported five cases with 100% technical success, no major complications, 60% partial response, and 40% progressive disease at 1 month follow-up [27].

#### 3.1.3. Cryoablation

Most of the current literature is focused on Radiofrequency ablation (RFA). However, this technique suffers from the “heat-sink” effect limitation. Furthermore, RFA and MWA have the inherent risk of thermal damage to proximal vital anatomical structures. In the setting of dangerous localizations, cryoablation can be a valid alternative [28].

Cryoablation is based on a cycle of freezing and thawing that causes intra- and extracellular ice crystal formation, damage to the cell membrane, and cell necrosis due to dehydration and changes in osmotic pressure [9]. The cooling mechanism of the ablation probe is based on the Joule–Thomson effect; the probe tip reaches temperatures as low as −160 °C thanks to the sudden expansion of high-pressure argon in a small chamber located in the probe tip. After the first phase of cooling and the creation of the ice ball, the tissue gradually thaws to 0 °C, and then a second freezing process starts; repositioning of the probes is possible if necessary. Tumors less than 3 cm in size may be treated with a single probe but larger tumors may require placement of multiple probes or sequential treatments [15]. A safety margin of at least 5 mm is recommended.

The major advantages of cryoablation are the visibility of the ice ball on ultrasound (US), computed tomography (CT), and magnetic resonance imaging (MR), the lower sensitivity to heat-sink effect compared to RFA [29], and the analgesic property of cold energy, which is associated with reduced intra- and postprocedural pain [30,31,32,33].

Cryoshock is a peculiar, extremely rare, and potentially life-threatening complication of cryoablation (0.3 to 2.0% of patients [34,35]). It is a cytokine-mediated biological process, clinically presented with disseminated intravascular coagulation and multi-organ failure [36].

As for MWA, the available data on pancreatic cancer cryoablation is very poor in literature.

There are only three papers that reported pancreatic cryoablation; two of them enrolled patients with stage 4 pancreatic cancer [32,37], and the third study used a combination of cryoablation and radioactive iodine-125 treatment [38]. The most interesting experience is that of Niu et al. [32], who compared four groups of patients with metastatic pancreatic cancer (stage 4): 22 patients underwent chemotherapy alone, 36 patients underwent cryotherapy alone, 17 patients underwent immunotherapy alone, and 31 patients underwent a combination of cryoablation and immunotherapy (cryoimmunotherapy). Median overall survival (OS) was significantly higher in the cryoimmunotherapy (13 mo) group when compared to the cryotherapy (7 mo), immunotherapy (5 mo), and chemotherapy (3.5 mo) groups. Regarding the analgesic property of cryoablation, it is interesting to note that Niu et al. reported at least a 50% decrease in pain score in 84% of the patients and a 50% decrease in analgesic consumption in 69% of the patients.

### 3.2. Irreversible Electroporation

IRE is a novel non-thermal ablation technology, first introduced by Davalos et al. in 2005 [39]; IRE is based on the application of pulsatile and targeted high-voltage electric energy [40] that alters the current potential of the cellular membrane, leading to permanent nanopore within the lipid bilayer membrane. This membranous disruption results in a loss of homeostasis and thus apoptosis and cell death.

Since extracellular matrix structures are typically preserved and the lack of the heat-sink effect, IRE is an attractive alternative for vascular invasive LAPCs, which remain unresectable after neoadjuvant chemo(radio)therapy. IRE treatment is possible for tumor sizes up to 5cm [15], but Narayanan et al. demonstrated that overall survival after IRE treatment is better in patients with a tumor size less than 3 cm [41].

Table 2 shows the main absolute and relative IRE contraindications.

IRE is typically performed under general anesthesia with a full neuromuscular blockade to avoid muscular contraction caused by high-voltage electricity [42,43]. Gastric emptying by nasogastric tube and cardiac monitoring are also required [44]. The purpose of cardiac synchronization are to guarantee the IRE pulse delivery during the absolute refractory period of the cardiac cycle to avoid ventricular tachycardia [45].

Because of the possibility of adjacent tissue swelling after IRE, it is highly recommended to ensure biliary protection before the ablation in all the tumors causing biliary obstruction and in those in proximity to the common bile duct [46,47]. For the same reason, portal vein stenting should be performed in patients with a partially occluded portal vein [48].

The interventional radiologist should aim for a parallel placement of the 19G electrodes (maximum angulation 10°) for an interelectrode distance of about 20 mm (15–24 mm) and for a tumor-free margin of at least 5 mm [48,49], as shown in Figure 1.

Electrodes can be repositioned during the procedure to ensure a wider ablative area [41,46].

A contrast-enhanced CT scan is performed immediately after the procedure to confirm the correct ablation zone and to check for early complications. The ablated area appears hypodense, usually with a hyperdense peripheral rim caused by reactive hyperemia. It is interesting to note that several studies have demonstrated that the hypodense zone on CT significantly corresponds to the zone of cell death [50,51].

A systematic review by Moris et al. reported a median overall survival (OS) following IRE between 7 and 27 months. Complete remission is rare (16% of the patients), with a partial response rate of 38% [52].

He et al. compared the efficacy of IRE with other treatments [53] in a non-randomized trial, showing that IRE and neoadjuvant chemotherapy is superior to RFA and neoadjuvant chemotherapy.

In a meta-analysis by Ansari et al. [54], a 2% mortality rate (9/446 patients) was reported. The severe complication rate ranged from 0% to 24% [55,56], while the minor complication rate was reported to be between 10% and 62% [41,57,58]. Unlike thermal ablation, the most common complications of IRE are nausea, vomiting, loss of appetite, and gastroparesis. Cholangitis, biloma formation, and severe pancreatitis are rare but may be observed after the procedure. Vascular complications, such as SMA obstruction and pseudoaneurysm, were also rarely reported.

### 3.3. Synergic Effect of Ablative Therapies and Immunotherapy; a Future Perspective

There has been a growing interest in recent years about the synergic effects of immunotherapy and locoregional strategies, but few data are available in the literature.

From an immunological point of view, the necrosis caused by the ablation techniques increases the antigen presentation by dendritic cells (APC), the serum cytokines level, the CTLA-4 cascade, and the activation of the T-cell response [59,60,61]. This immunological activation has both local and systemic (i.e., Abscopal) effects [60,62,63,64], thereby paving the way for an expanded role of these procedures as stimulants to the immune system.

On the other hand, classical systemic immunotherapy has only limited efficacy against pancreatic ductal adenocarcinoma (PDAC) due to the presence of an immunosuppressive tumor-associated stroma [65]. Therefore, the rationale of the early studies was that ablative therapies could destroy the pancreatic immunosuppressive microenvironment, thereby leading to a greater response to systemic immunotherapy.

Timmer et al. reported the most interesting studies in this field applied to pancreatic cancer [66].

Zhao et al. used a mouse model of PDAC; they demonstrated that the association of IRE and systemic anti-PD1 treatment promotes CD8+ T-cell infiltration and increases overall survival when compared to both IRE and anti-PD1 as monotherapy [65].

Narayanan et al. also used a mouse model of PDAC; in their experience, IRE was combined with systemic anti-PD1 and an intra-tumoral TLR-7 agonist. This triple strategy improved local response when compared to IRE alone and promoted regression of untreated concomitant metastases [67].

These encouraging results have been applied to preliminary human studies; it has been suggested that IRE combined with NK cells [68,69] or allogenic Vγ9Vδ2 T-cell infusion [70] could have life-prolonging effects. This is the first clinical study combining IRE with γδ T cells in LAPC patients (n = 62). Patients were randomized to receive IRE alone (n = 32) or IRE and γδ T cells (n = 30); the latter group achieved longer progression-free (8.5 vs. 11 months) and overall survival (11 vs. 14.5 months).

The PANFIRE-III trial will combine IRE, systemic anti-PD1, and an intra-tumoral TLR-9 agonist in metastasized PDAC patients (NCT04612530).

It is undeniable that future studies are necessary to obtain the maximum synergy between immunotherapy and interventional radiology, but the initial results obtained are encouraging.

## 4. Pancreatic Liver Metastases

More than half of patients with pancreatic cancer have metastatic disease at the time of diagnosis, with a poor 5-year survival rate of around 3% [9,71]. Most patients are unresectable [72], so the only therapeutic option available to them is chemotherapy [73] (i.e., FOLFIRINOX) with the possibility of severe toxicity, such as neutropenia, diarrhea, thrombocytopenia, neuropathies, and fever [74]. In patients with liver-dominant metastatic disease, interventional locoregional treatments could provide a chemotherapy holiday and offer a survival benefit.

### 4.1. Chemoembolization in Pancreatic Liver Metastases

Transarterial therapies are an essential tool for the management of primary and secondary liver malignancies.

Transarterial chemoembolization (TACE) is an endovascular technique in which an embolic agent is delivered in association with a chemotherapeutic drug. The usefulness of TACE is because the drug is significantly more locally concentrated, with a consequent sparing of the remaining liver, and with a reduction of systemic side effects. The embolizing effect reduces blood flow within the lesion, leading to ischaemia, thereby prolonging the dwell time of the chemotherapic agent in the metastasis and further decreasing the systemic side effects of the drug [75,76,77].

There are different types of TACE. The most common is conventional TACE (c-TACE), in which the cytotoxic drug is delivered with lipiodol to the tumor and the arterial vessel is subsequently embolized with sponge, and drug-eluting bead TACE (DEB-TACE), in which microspheres carrying the drug are delivered with sustained drug delivery, followed by embolization [78].

TACE is primarily used in the treatment of unresectable hepatocellular carcinoma and colon cancer liver metastases, but there is little data on TACE in metastatic pancreatic cancer.

Vogl et al. [79] published a study in which 112 patients with pancreatic cancer liver metastases underwent TACE performed with a combination of cisplatin, mitomycin C, and gemcitabine followed by iodized oil and embolization microspheres (50 μm) until reaching vascular stasis, achieving an overall survival of 19.2 months and 5-year survival rate of 50%.

Azizi et al. [80] retrospectively analyzed 32 patients who were treated with the same TACE technique as in Vogl et al.’s study, achieving a median overall survival of 16 months.

Another study by Sun et al. [81] showed, in a group of 27 patients, that TACE for pancreatic cancer liver metastases improved QoL and the median overall survival (OS) (23 months).

### 4.2. Ablative Treatment in Pancreatic Liver Metastases

Liver thermal ablation is routinely used to treat both primary and secondary unresectable liver malignancies [82], leading to necrosis of the lesion with results similar to surgery [83]. There are many studies showing thermal ablation as a treatment option in patients with pancreatic cancer liver metastases; most of them use RFA as an ablative method.

A study published by Park et al. [84] retrospectively analyzed 34 patients with pancreatic cancer liver metastases who underwent thermal ablation with RFA. 18 patients (58%) underwent a second ablation due to recurrence of the metastatic disease, and 16 of them showed a good hepatic disease control; nine patients underwent a third ablation due to recurrence, while one patient underwent a fourth ablation. The median overall survival was 15 months. The study reveals that lesions less than 2 cm in size and with well differentiated histology are good prognostic factors for overall survival.

Hua et al. [85] published a retrospective study of 102 patients with pancreatic cancer liver metastases who underwent liver RFA, showing a median overall survival of 11.4 months.

Lee et al. [86] retrospectively compared 60 patients treated with RFA for pancreatic liver metastases and 66 treated with systemic therapy. The median overall survival was 3 months higher in the RFA group (12 vs. 9 months).

## 5. Final Considerations

Interventional radiology, which already plays a key role in the diagnosis of pancreatic cancer (through US- or CT-guided biopsies and cytologic brushing) and in the palliation of obstructive jaundice (through placement of biliary drainage or stents), is also gaining a central role in management of loco-regional ablative therapies and in the treatment of liver metastases.

As already mentioned, all loco-regional ablative techniques share the same indications, i.e., patients with LAPC that do not respond to neoadjuvant chemoradiotherapy, and patients unfit for surgery [8].

RFA and IRE are the most studied methods in terms of safety and efficacy [10]. Regarding efficacy, RFA has a median OS of 25.6 mo [19] and a 1-year OS of 46% [21]; a systematic review on IRE reported a median OS between 7mo and 27 mo [52]. A comparison between the two methods is difficult due to the lack of data in the literature and due to the heterogeneity of the study samples. Despite this, in a non-randomized study, He et al. demonstrated the superiority of IRE over RFA in terms of efficacy [53]. These preliminary results are being improved through the synergy of these methods with immunotherapy [59,60,61,62,63,64,65,66,67,68,69,70].

Regarding safety, the rate of complications and mortality in the literature is similar for IRE and RFA [22,23,41,54,55,56,57,58].

Few data are available for cryoablation and MWA. A preliminary study on cryoablation shows an OS of cryoablation and immunotherapy (13 mo) greater than cryoablation (7 mo), immunotherapy (5 mo), and chemotherapy (3.5 mo) as monotherapies [32]; although this technique is less studied, it has the undoubted advantages of the analgesic effect and the ice ball observability in real time [32]. It should also be emphasized that cryoablation is the only thermal ablative technique that does not induce denaturation of cellular proteins but determines the release of non-denatured antigens into the circulation; this could induce a more consistent post-ablative immune response against the tumor than the other techniques [59].

Regarding the MWA, we found only four results in the literature [24,25,26,27] and each with a small sample of patients; the hypothetical advantages of MWA are the higher intratumoral temperature and the larger ablated area due to a lower heat sink effect.

More than half of patients with pancreatic cancer have metastatic disease. The two experiences of Vogl et al. [79] and Azizi et al. [80] show the effects of TACE on survival (in terms of OS and 5-year survival rate) of end-stage patients; the study by Sun [81] also reveals the improvement in the quality of life after this treatment. Future studies will allow us to understand whether the different TACE techniques (e.g., b-TACE, DSM-TACE), which are already widespread in the treatment of HCC, can be used as an additional weapon in the treatment of multimetastatic patients.

On the other hand, studies such as the one by Lee et al. [86] demonstrate that patients with liver metastases treated with RFA have a higher survival rate than ones treated with systemic therapy alone (median overall survival of 12 vs. 9 months). Therefore, further studies are underway to understand the advantages (in terms of safety and efficacy) of other ablative techniques (e.g., MWA, cryoablation, and electroporation) applied to liver metastases from pancreatic cancer.

In conclusion, the choice of the best ablative or chemoembolization therapy should be made by a highly specialized multidisciplinary team. The choice primarily depends on the personal preference of the operator, on the materials available in the various institutes, and, only secondarily, on the data reported in the literature. Future research efforts in this and other fields (such as nanotechnology [87]) will improve the prognosis of patients with this deadly disease.

## Figures and Tables

**Figure 1 life-13-00835-f001:**
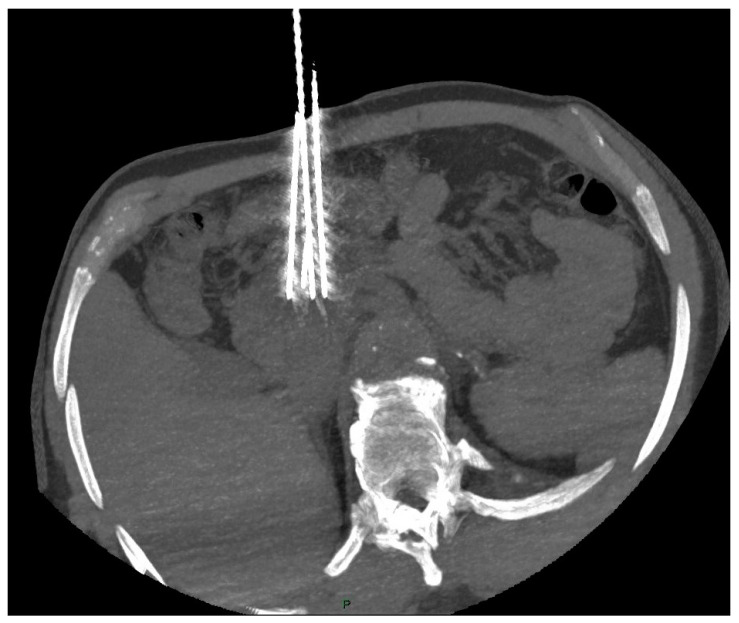
CT scan shows correct placement of the 4 electrodes (19 G) at the periphery of the pancreatic head tumor (interelectrode distance < 20 mm and angulation < 10°).

**Table 1 life-13-00835-t001:** Definition of resectability according to NCCS guidelines.

RESECTABILITY STATUS	ARTERIAL	VENOUS
**RESECTABLE**	No arterial tumor contact [celiac axis (CA), superior mesenteric artery (SMA), or common hepatic artery (CHA)]	No tumor contact with the superior mesenteric vein (SMV) or portal vein (PV) or ≤180° contact without vein contour irregularity
**BORDERLINE** **RESECTABLE** ^A^	Pancreatic head/uncinate process: Solid tumor contact with CHA without extension to CA or hepatic artery bifurcation allowing for safe and complete resection and reconstruction Solid tumor contact with the SMA of ≤180° Solid tumor contact with the variant arterial anatomy (ex: accessory right hepatic artery, replaced right hepatic artery, replaced CHA, and the origin of replaced or accessory artery); the presence and degree of tumor contact should be noted if present, as it may affect surgical planning. Pancreatic body/tail: Solid tumor contact with the CA of ≤180° Solid tumor contact with the CA of >180° without involvement of the aorta and with intact and uninvolved gastroduodenal artery thereby permitting a modified Appleby procedure (some panel members prefer these criteria to be in the locally advanced category)	Solid tumor contact with SMV or PV of >180°, contact of ≤180° with contour irregularity of the vein or thrombosis of the vein but with suitable vessel proximal and distal to the site of involvement allowing for safe and complete resection and vein reconstruction. Solid tumor contact with the inferior vena cava (IVC)
**LOCALLY ADVANCED** ^A,B^	Pancreatic head/uncinate process: Solid tumor contact with the SMA > 180° Solid tumor contact with the CA > 180° Pancreatic body/tail: Solid tumor contact of >180° with the SMA or CA Solid tumor contact with the CA and aortic involvement	

^A^ Solid tumor contact may be replaced with increased hazy density/stranding of the fat surrounding the peri-pancreatic vessels (typically seen following neoadjuvant therapy); this finding should be reported on the staging and follow-up scans. ^B^ Distant metastasis (including non-regional lymph node metastasis), regardless of anatomic resectability, implies disease that should not be treated with upfront resection.

**Table 2 life-13-00835-t002:** Absolute and relative contraindications to IRE.

**ABSOLUTE CONTRAINDICATIONS**	- Pervasive involvement of the duodenum - History of epilepsy - Cardiovascular diseases - history of ventricular arrhythmias - implanted cardiac stimulation devices - congestive heart failure with NYHA class > 2 - uncontrolled hypertension
**RELATIVE CONTRAINDICATIONS**	- Atrial fibrillation - Coronary artery diseases - Combined severe stenosis of the common hepatic artery and main portal vein branch - Metallix foreign object in the ablation zone - Impeded liver function - Irreversible disorders - Uncontrolled infections - Chemo- or immunotherapy in the last 4 weeks

## Data Availability

The data that support the findings of this study are openly available in the three main electronic bibliographic databases (PubMed, BioMed Central, and Scopus).

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
