# Peer review of "Interventional Radiology in the Treatment of Pancreatic Adenocarcinoma: Present and Future Perspectives"

_life, 2023, doi:10.3390/life13030835_

Round 1

Reviewer 1 Report

this is an excellent narrative review on interventional radiology in the treatment of pancreatic adenocarcinoma. I feel that it is quite comprehensive and well written, and in my opinion it deserves dissemination to the public domain. I only wonder why the Authors chose 'Life' as a target journal, maybe this review can be of interest for a more narrow audience.

I have one minor comment. I understand that this is a narrative review - and the Authors clarify this in the Introduction - but can some details on the literature research be provided?

Author Response

Thanks for your comments.

We selected this journal because we believe that interventional radiology is going to take a central role in the fight against cancer and therefore we think that topics such ours will be of interest to a wide audience.

We agree with you. We added a brief paragraph briefly explaining how the literature research was provided.

Reviewer 2 Report

The authors review Interventional Radiology in the treatment of pancreatic adenocarcinoma.

The review is so interesting and the manuscript is well written, however, I have some concerns to discuss.

-Please describe your past studies and compare other studies in Table.

-What is the novelty of the current review?

-How should we treat pancreatic adenocarcinoma in conclusion?

Author Response

Thanks for your comments. We think that the major novelty of our narrative review is the synergistic effect of immunotherapy and interventional radiology. As you suggested, we modified the conclusions to better underline our “take home message”: the best treatment option can only be selected by a highly specialized multidisciplinary team, based on patient and tumor characteristics, preference/skills/experiences of the operators, devices available in the institution, literature, and patient preference.

Reviewer 3 Report

The article "Interventional Radiology in the treatment of pancreatic adenocarcinoma: present and future perspectives" by Ernesto Punzi et al.  provides an interesting overview of the main approaches for the treatment of pancreatic adenocarcinoma.

Pancreatic cancer is the fourth leading cause of cancer death in the United States and will become the second by 2030. Pancreatic ductal adenocarcinoma of the pancreas (PDAC) accounts for more than 90% of all pancreatic cancers. PDAC is a devastating neoplasm with an extremely poor prognosis. The median survival duration is less than 6 months after diagnosis, and the 5-year overall survival rate is less than 7%. Pancreatic adenocarcinoma is the 12th most common malignancy and the 7th leading cause of cancer mortality globally. Most patients present with late symptoms with locally extensive or metastatic disease. The aggressive nature, late presentation, and lack of effective therapies contribute to the poor prognosis. The absence of specific symptoms and the aggressiveness of the disease are the two main limitations to early diagnosis. As is the case in most clinical disciplines today, a multidisciplinary approach is essential for the management of PDAC because of the need for primary prevention, the importance of early diagnosis, and the complexity of treatment since, even in combination with radiotherapy, traditional therapeutic strategies have not prolonged 5-year survival rates (less than 30%). To date, surgical resection with curative intent remains the only treatment option. However, at initial presentation, only 15-20% of patients have a surgically resectable tumor and, in addition, 45-50% of patients have overtly metastatic disease. The remaining 25-30% of patients have borderline resectable pancreatic cancer (BRPAC) or locally advanced pancreatic cancer (LPAC).  This condition opens up the concept of pancreatic tumor resectability by dividing pancreatic tumor progression into BRPAC, LPAC, or advanced pancreatic cancer (APAC). As a general principle, the estimation of resectability of PDAC should always be guided by the ability to obtain negative resection margins (R0 margins). BRPAC has been defined as a condition that encompasses a spectrum of patients ranging from "resectable" disease to LPAC. For these patients, a resection with a microscopically positive (R1) margin is considered relatively more likely, mainly because of the relationship between the primary pancreatic tumor and the surrounding blood vessels. LPAC is defined as a tumor condition involving the celiac trunk, or having a tumor-artery interface >180° and/or SMV/PV involvement without reconstruction options. APAC/LPAC is finally defined as an unresectable pancreatic tumor with metastasis.

PDAC grows rapidly, metastasizes early, and is generally accompanied by significant resistance to adjuvant therapeutic strategies. Few patients meet the criteria for curative resection because most are diagnosed with advanced disease. Palliative chemotherapy and radiotherapy remain the only treatment options for most patients with inoperable PDAC. These therapies offer only limited efficacy because PDAC recurs frequently, has multidrug resistance and low radiosensitivity. Approximately 20% of patients with acute pancreatitis develop complications that require intervention. These can be classified into vascular and nonvascular complications: Nonvascular complications include collections, bowel complications, and pancreatic fistulas. Vascular complications include peripancreatic arterial and venous pseudoaneurysms, venous thrombosis, and arteriovenous malformations (AVMs). This situation poses an unmet therapeutic challenge for the development of new treatments. New agents and strategies are needed for the management of inoperable PDAC.

Interventional treatments administered locally to directly damage cancerous tissue have been shown to be associated with reduced infection rates, rapid recovery, and shorter hospital stay. Local tumor ablation and embolization, e.g., irreversible electroporation (IRE) and image-guided nanoelectroablation, radiofrequency ablation (RFA), microwave ablation, cryoablation, transarterial chemoembolization and/or radioembolization, and endoscopic ultrasound (EUS)-guided administration and therapy, have been reported as options for treating patients with advanced PDAC. However, these interventional treatments are more likely to be used to help prevent or relieve cancer symptoms and are often used in conjunction with other types of treatments.

A challenging new approach is interventional nanotherapy, which incorporates the use of NPs with interventional treatments to improve antitumor efficacy through drug delivery, enhancement of chemotherapy and radiosensitivity, increased tumor uptake of therapeutic agents, and mediation of thermal effects, thereby potentially improving outcomes for patients with PDAC.

Research in nanomedicine and nanotechnology for cancer diagnosis and treatment has undergone unprecedented expansion in recent years. However, major obstacles exist for NP-mediated cancer therapeutics, especially in PDAC. The hypovascular nature of PDAC may prevent NP deposition in the tumor after systemic administration, and most NPs localize predominantly in the mononuclear phagocytic system, with a relatively poor uptake ratio between tumor and surrounding organs. Image guidance combined with minimally invasive interventional procedures can help overcome these barriers to poor NP delivery in PDAC. Interventional treatments allow regional drug delivery, targeted vascular embolization, direct tumor ablation, and the ability to disrupt the stromal barrier of PDAC. Interventional treatments also have potentially fewer complications, faster recovery, and lower costs than conventional therapies.

Cancer nanoteranostics, with the purpose of combining imaging and therapy using nanotechnology, is in great expandion in biomedical research due to the unique biological properties of nanoscale effect and the multifunctional capabilities of nanomaterials. Currently, enormous research efforts result in many advanced nanological platforms. Multimodal treatments have been integrated into single nanoparticle (NP) systems, with cytotoxic agents and/or tumor targets loaded on the surface, trapped within or dissolved in the NP matrix to achieve preferential accumulation within tumor cells, overcome drug resistance, and exert unique functions such as photothermal conversion, radiosensitization, drug transport, and contrast for imaging. Image-guided interventional techniques in combination with multifunctional NPs may offer some solutions for specific delivery to the tumor target. Interventional oncology encompasses a variety of minimally invasive, real-time image-guided procedures for the diagnosis and treatment of cancer and is generally associated with fewer complications, faster recovery times, and lower costs than surgery. Interventional techniques allow regional drug delivery, targeted vascular embolization, and direct tumor ablation.

Authors are requested to briefly report, along with the procedures described, the recent application of interventional radiology procedures in conjunction with the use of nanoparticles.

Author Response

Thanks for your revision and well-explained point of view. We agree with you. As suggested, we briefly mentioned nanoparticles technology in the conclusion. We believe that a further description of this fascinating technology is beyond the scope of our article and beyond our competences; by quoting a recent article on this field, the most interested readers can explore the topic independently (The article cited is ''Caputo D, Pozzi D, Farolfi T, Passa R, Coppola R, Caracciolo G. Nanotechnology and pancreatic cancer management: State of the art and further perspectives. World J Gastrointest Oncol. 2021 Apr 15;13(4):231-237'').

Round 2

Reviewer 2 Report

The authors replied well, so the manuscript is suitable for publication.